# Pelvic floor disorders and associated factors among women in sub-Saharan Africa: A systematic review and meta-analysis protocol

**Atoma Negera** [1]*, **Midekso Sento**[2], **Geleta Nenko**[3], **Gamachis Firdisa**[4], **Jira Waqoya**[5,6], **Samuel Negera**[7], **Bilisumamulifna Tefera**[8]

**1** Public health course unit, Nursing Department, College of Health Science, Mattu University, Mattu town, Southwest Ethiopia, **2** Anatomy course unit, Biomedical Department, Adama Hospital Medical College, Adama town, Eastern Ethiopia, **3** Health Informatics Department, College of Health Science, Mattu University, Mattu town, Southwest Ethiopia, **4** Maternity and Reproductive health course unit, Nursing Department, College of Health Science, Mattu University, Mattu town, Southwest Ethiopia, **5** Public Health Department, Institute of Health, Wallaga University, Nekemte town, Southwest Ethiopia, **6** The Hong Kong Polytechnic University, Hong Kong, SAR, **7** Medical Laboratory Department, College of Health Science, Oda Bultum University, Chiro town, East Ethiopia, **8** Public Health Department, College of Health Science, Mattu University, Mattu town, Southwest Ethiopia

* atomanegera@gmail.com

## Abstract

### Background

Pelvic floor disorders (PFDs) are a group of conditions caused by injured or weakened pelvic muscles, ligaments, connective tissues, and nerves that support or hold pelvic organs in place so they can function correctly. Common PFDs are pelvic organ prolapse (POP), urinary incontinence (UI), and faecal incontinence (FI). A preliminary search on the subject within the last decade identified no review protocol or systematic review, despite a significant percentage of women in SSA suffering from it.

### Methods and analysis

A comprehensive literature search will be gathered from electronic databases such as PubMed, Embase, Hinari, Cochrane Library, African Journals Online (AJOL), and Google Scholar. The protocol followed the Preferred Reporting Items for Systematic Reviews and Meta-analyses for Protocol (PRISMA-P) guideline. All studies conducted in sub-Saharan African countries will be included regardless of their study design as long as these studies report the magnitude of the problem under study. Joanna Briggs Institute's (JBI) appraisal checklist will be used to assess the quality of individual studies. Heterogeneity will be checked using Cochrane Q test statistics and $I^2$ test statistics, and a random-effects model will be employed to estimate the pooled prevalence of PFDs and its associated factors.

**Data availability statement:** All relevant data from this study will be made available upon study completion.

**Funding:** The author(s) received no specific funding for this work.

**Competing interests:** The authors have declared that no competing interests exist.

## Results

The present study will estimate the pooled prevalence of pelvic floor disorders and their associated factors in sub-Saharan Africa countries.

## Systematic review registration

This review was registered on PROSPERO with registration number CRD42024578550.

## Introduction

Pelvic floor disorders (PFDs) are a group of conditions caused by injured or weakened pelvic muscles, ligaments, connective tissues, and nerves that support or hold pelvic organs in place so they can function correctly. Common PFDs are pelvic organ prolapse (POP), urinary incontinence (UI), and faecal incontinence (FI) [1,2]. About one-third of adult women in the United States are affected with at least one pelvic floor disorder, and several studies predicted that the number of cases of pelvic floor disorders would rise significantly [3,4].

Even though a significant percentage of women in SSA countries suffer from PFDs, there is little knowledge about PFDs among women in developing countries like SSA [5]. According to a study conducted in developing countries, the pooled prevalence of pelvic Organ prolapse was 19.7% (range 3.4–56.4%), urinary incontinence was 28.7% (range 5.2–70.8%), and fecal incontinence was 6.9% (range 5.3–41.0%) [6]. Furthermore, another study conducted in low- and middle-income countries (LMICs) found that the pooled prevalence of PFD was 25% [7]. In SSA, where there is a social stigma, a lack of access to healthcare, malnourishment, higher comorbid diseases, and a lack of awareness about the condition and its management [6], the true prevalence of PFD may be higher than this. Consequently, we believe that PFD is typically underreported and undiagnosed in SSA.

Previous research has shown that pelvic floor diseases are prevalent and closely linked to the following factors: female gender, higher BMI, aging, higher parity, menopause, hysterectomy, constipation, instrumental delivery, and early age at first birth, prolonged second-stage labor, prolonged heavy manual labor, and persistent coughing. All these factors lead to mechanical stressors, which raise the risk of PFDs in women [3,5,6,8]. Co-morbid conditions and undernutrition significantly increase this risk [6].

PFDs are a significant public health issue that adversely affects the lives of millions of women in several ways, such as physical, psychological, sexual, occupational, behavioral, and social domains [9,10]. Despite having a negative impact on women's quality of life, the issue received little attention. By estimating the pooled prevalence of pelvic floor disorders across sub-Saharan African nations, this study contributes to the development of suitable care and preventative strategies.

To avoid duplication, a preliminary search of similar reviews or review protocols was conducted in the databases of Google Scholar, MEDLINE, and The International Prospective Register of Systematic Reviews (PROSPERO). Despite the fact that a significant percentage of women in the SSA suffer from it, no review protocol or systematic review was identified on the subject that has been published in the SSA within the last decade.

## Objectives of the review

The primary objective of this review is to determine the pooled magnitude of pelvic floor disorder in sub-Saharan Africa.

The secondary objective of this review is to identify factors associated with pelvic floor disorder in sub-Saharan Africa.

## Method and materials

### Review registration and report

This review protocol has been registered in the International Prospective Register of Systematic Reviews (PROSPERO) with registration number CRD42024578550. The study will follow the Preferred Reporting Items for Systematic Reviews and Meta-Analysis Protocols (PRISMA-P) checklist [11]. The checklist is presented in the supporting information [S1 Appendix]. The review will commence between September 1 and 20, 2024, and necessary amendments will be published along with the results of the systematic review and meta-analysis.

### Data sources and Search strategy

An electronic database search for published articles and gray literature will be done from the following databases: PubMed, Embase, Hinari, Cochrane Library, African Journals Online (AJOL), and Google Scholar. Additional searches from references to identified literature will be performed. All research articles containing information on the prevalence of symptomatic PFDs published after 2000 years and until August 20, 2024, will be retrieved. The search strategy included a combination of subject terms and free text terms. The Medical Subject Headings (MeSH) terms included "pelvic floor disorders", "pelvic organ prolapse", "genital prolapse", "uterine prolapse", "urinary incontinence", "stress urinary incontinence", "faecal incontinence", "anal incontinence", "prevalence", "magnitude", "sub-Saharan Countries", "resource-poor", "resource-limit", "low-income", "lower-middle-income countries", "developing countries". These terms were combined with Boolean operators ('OR', 'AND', and "NOT"). A detailed search strategy is presented in the supporting information [S2 Appendix].

### Eligibility criteria

**Inclusion criteria.** Observational studies that reported prevalence of women with PFDs using validated data collection tools and conducted in sub-Saharan Africa will be included in the review. However, if multiple publications are generated from the same data with the same outcome, only the most relevant publications will be included. If the findings of the study were reported in languages other than English, online translation software will be used to extract the necessary data.

**Exclusion criteria.** Editorials, opinion articles, letters, narrative or systematic reviews, brief communications, conference abstracts, and posters will be excluded. Studies that did not assess the prevalence of PFDs (studies that evaluate treatments for PFDs, the quality of life of women with PFDs, etc.) will be excluded.

### Study selection and data extraction

All identified citations will be collated and uploaded into the Covidence online systematic review tool as recommended by the Cochrane Handbook [12]. This tool is designed to help reviewers remove duplicates, screen abstracts, and the full texts of identified articles, assess the risk of bias in included articles, and perform data extraction. Abstracts of the relevant full texts will be assessed for eligibility by two reviewers (AN and MS) independently. Full-text articles for the selected titles will be further reviewed independently by these reviewers. Disagreements will be resolved by consensus where possible or by a third reviewer (GF) as needed. Two of the authors (GF and SN) will extract data independently using a customized data extraction form in Covidence [13]. All papers finally selected will be cross-checked by two other authors (GN and BM). Any disagreement on a particular paper will be resolved by discussion before inclusion in the study. Data extracted will include author(s), journal, year

of publication, type of the study, study design, country of origin, rationale of the study, study population, sample size, outcomes of the study, key findings related to the review, limitations, and recommendations.

### Risk-of-bias and quality assessment

A risk-of-bias tool will be assessed using the Joanna Briggs Institute (JBI) appraisal checklist developed explicitly for systematic reviews of prevalence studies [14] and is presented in the supporting information [S3 Appendix]. Two review authors (AN and JW) will assess the risk of bias independently and assign each study a JBI score ranging from 1 to 10, with higher scores indicating higher quality; inconsistencies will be identified and resolved through discussion involving a third author where necessary.

### Statistical analysis

After data extraction into an Excel spreadsheet, data will be exported to Stata for windows version 15 statistical software for analysis statistical software for analysis [15]. The pooled prevalence of included studies will be calculated. Summary statistics of individual studies will be presented in a graph. Cochrane's Q statistic, $I^2$, and p-value will be used to check the heterogeneity of the study's outcomes. $I^2$ of 25%, 50%, and 75% will be used as an indicator for low, moderate, and high heterogeneity, respectively [16]; forest plots will be used to visualize heterogeneity. If there is moderate to high heterogeneity, random effect meta-analysis will be employed. Meta-regression will be used to identify the source of heterogeneity. A statistically significant result from meta-regression will be declared as a source of heterogeneity. Meta-regression will be used to investigate pooled prevalence differences between PFD subtypes, economic level of countries (low-income vs. middle-income countries), sample size of the study (large vs. small sample size), sampling techniques will be used (random vs. convenience), and publication year (2010 to recent years vs. years before 2010). We will assess the presence of publication bias using a funnel plot; a statistical significance result from Egger and Begg tests will also be used as an indicator of publication bias. Leaveone-out sensitivity analysis will be performed to assess a single study effect.

### Operational definitions

**Pelvic floor disorder (PFD)** was evaluated based on whether the woman stated one or more of the indicators of urine leakage (UI), fecal incontinence (FI), and pelvic organ descent. For the purposes of the study, women with at least one UI, FI, or pelvic organ prolapse were assumed to have pelvic floor dysfunction [17].

**Urine incontinency (UI)** considered if the woman complains of an uncontrolled loss of urine related to both urge and stress urine leakage [17].

**Anal incontinence (AI)** was deemed if the woman has responded with an unintentional loss of solid or liquid feces at least for a month for the past year [17].

**Pelvic organ prolapse (POP)** A pelvic organ prolapse was evaluated when the woman reported a feeling of bulging, pressure, or when something seemed to be coming down through the vagina and a visible mass was bulging from the vaginal canal [17].

### Ethics and dissemination of the results

Ethics committee approval or written informed consent will not be required to conduct this systematic review, and meta-analysis for the review will be entirely based on published data. The result of this study will be submitted to a peer-reviewed journal, and it will also be presented at relevant research conferences.

**Strengths and Limitations of the study.** This systematic review will address a global gap by providing the pooled prevalence of PFDs. The review will identify factors associated with the pelvic floor disorder. This review protocol follows the Preferred Reporting Items for Systematic Review and Meta-analysis Protocols (PRISMA-P) guidelines with transparency regarding the methods and processes that will be used. Among the limitations that could be anticipated, the studies found could have a small sample size. There could also be heterogeneity in relation to participants and intervention.

## Conclusions

This systematic review will determine the magnitude of pelvic floor disorder in sub-Saharan Africa with a detailed summary of factors associated with pelvic floor disorder to improve women's health related to PFD. The findings of this systematic review and meta-analysis can help strengthen evidence-based knowledge and contribute to recommendations for further research.

## Supporting information

**S1 Appendix. PRISMA-P checklist.**
(DOCX)

**S2 Appendix. Searching strategy**
(DOCX)

**S3 Appendix. JBI Critical appraisal tool.**
(DOCX)

## Acknowledgement

None

## Author contributions

**Conceptualization:** Atoma Negera, Midekso Sento.

**Data curation:** Geleta Nenko.

**Formal analysis:** Geleta Nenko.

**Investigation:** Samuel Negera.

**Methodology:** Atoma Negera.

**Project administration:** Gamachis Firdisa.

**Resources:** Jira Waqoya, Samuel Negera.

**Supervision:** Atoma Negera, Gamachis Firdisa, Jira Waqoya.

**Validation:** Bilisumamulifna Tefera.

**Writing – original draft:** Atoma Negera, Midekso Sento, Gamachis Firdisa.

**Writing – review & editing:** Atoma Negera, Geleta Nenko, Jira Waqoya, Samuel Negera, Bilisumamulifna Tefera.

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
