## [Decision Letter · Decision Letter 0]

6 Jan 2025

PONE-D-24-38448Pelvic floor disorders and associated factors among women in sub-Saharan Africa: A Systematic Review and Meta-analysis Protocol

PLOS ONE

Dear Dr. Negera,

Thank you for submitting your manuscript to PLOS ONE. After careful consideration, we feel that it has merit but does not fully meet PLOS ONE’s publication criteria as it currently stands. Therefore, we invite you to submit a revised version of the manuscript that addresses the points raised during the review process.

The manuscript has been evaluated by four reviewers, and their comments are available below.

The reviewers have raised a number of concerns that need attention. They request additional information on methodological aspects of the study, and reconsideration of the inclusion criteria.

Could you please revise the manuscript to carefully address the concerns raised?

We look forward to receiving your revised manuscript.

Kind regards,

Helen Howard

Staff Editor

PLOS ONE

Journal Requirements:

Reviewers' comments:

Reviewer's Responses to Questions

**Comments to the Author**

1. Does the manuscript provide a valid rationale for the proposed study, with clearly identified and justified research questions?

Reviewer #1: No

Reviewer #2: Yes

Reviewer #3: Yes

Reviewer #4: Yes

2. Is the protocol technically sound and planned in a manner that will lead to a meaningful outcome and allow testing the stated hypotheses?

Reviewer #1: Yes

Reviewer #2: Yes

Reviewer #3: Partly

Reviewer #4: Yes

3. Is the methodology feasible and described in sufficient detail to allow the work to be replicable?

Reviewer #1: Yes

Reviewer #2: Yes

Reviewer #3: Yes

Reviewer #4: Yes

4. Have the authors described where all data underlying the findings will be made available when the study is complete?

Reviewer #1: Yes

Reviewer #2: Yes

Reviewer #3: Yes

Reviewer #4: Yes

5. Is the manuscript presented in an intelligible fashion and written in standard English?

Reviewer #1: Yes

Reviewer #2: Yes

Reviewer #3: Yes

Reviewer #4: Yes

6. Review Comments to the Author

You may also provide optional suggestions and comments to authors that they might find helpful in planning their study.

Reviewer #1: The authors present a protocol for a systematic review of pelvic floor disorders and associated factors among women in sub-Saharan Africa. While this is an important topic for understanding the burden and determinants of pelvic floor disorders in this population, the rationale for publishing the protocol, as opposed to the actual study, is not clearly articulated. The protocol outlines a comprehensive approach to data collection and interpretation; however, the authors do not explain why the protocol itself warrants publication. Additionally, the aim and scope of this research might be better addressed through collaboration with the Global Burden of Disease study. Even if the authors wish to pursue this review by analyzing the literature, this approach does not necessarily justify the protocol's publication, as it neither significantly advances the literature nor provides a clear trajectory for future research. Moreover, the authors fail to present a comprehensive search strategy across relevant databases, which is crucial for conducting a systematic review. Given the stated goal of performing a thorough systematic search, it is imperative that the complete strategy be provided.

Reviewer #2: The review question is clear

The protocol is adequately developed based on international Rw guidelines

Tools such as Covidence helps for an efficient screening of the included studies

The methodological approach using PRISMA-P is adequate and clear as the review protocol.

How the research team will implement each step and who will be involved at each stage is well detailed

Joanna Bridge Institute's (JBI) appraisal should read as Joanna Briggs Institute

Reviewer #3: December 23rd, 2024

Dear Editor,

Dear Authors,

Thank you for affording me the valuable opportunity to review this manuscript.

This methodological study offers significant insights into the prevalence and risk factors of pelvic floor disorders (PFDs) in sub-Saharan Africa, an area where existing evidence remains limited. The findings are poised to make a substantial contribution to the field.

However, the inclusion criteria need to be reassessed.

Pelvic floor disorders (PFDs) can be evaluated through clinical examination or self-administered questionnaires. For instance, the Pelvic Floor Disability Index (PFDI-20) is a self-reported questionnaire that does not determine prevalence but quantifies symptom burden.

In such cases, the findings primarily elucidate symptomatic presentation rather than true prevalence, as exemplified in the reference study (Suemitsu T, Mikuni K, Matsui H, Suzuki M, Takahashi T. Prevalence and Risk Factors of Pelvic Floor Disorders After Delivery in Japanese Women Using the Pelvic Floor Distress Inventory: A Retrospective Cohort Study. Cureus. 2023;15(6):e40152. Published 2023 Jun 8. doi:10.7759/cureus.40152).

The inclusion criteria will inevitably vary depending on whether the study aims to determine prevalence or symptom burden. Conducting a preliminary evaluation of previous studies to clarify their assessment methods, the volume of relevant literature, and the reliability of these sources would be prudent.

Additionally, I suggest considering treatment and intervention strategies, including rehabilitation, as secondary evaluation outcomes. Identifying risk factors does not always equate to their mitigation. For example, the increased incidence of PFDs associated with instrumental delivery, obstetric anal sphincter injuries (OASIS), and epidural analgesia is a well-documented phenomenon worldwide, not exclusive to sub-Saharan Africa.

A review of this nature may yield findings similar to previous studies. Nevertheless, presenting potential solutions would substantially enhance the review's significance. Therefore, I recommend incorporating this aspect into the evaluation criteria.

I appreciate your consideration, and I look forward to your response.

Sincerely,

T.SUEMITSU

Reviewer #4: I would like to thank you for the opportunity to review the study protocol entitled "Pelvic Floor Disorders and Associated Factors Among Women in Sub-Saharan Africa: A Systematic Review and Meta-Analysis Protocol". This protocol raises considerations regarding the inclusion of studies for analysis, and I would like to make a few suggestions.

First, it is important to determine whether the studies selected for review have adequately addressed the prevalence of pelvic floor disorders. It is recommended that the included studies use validated assessment tools that are appropriate for the target population. The lack of validity may be a source of bias in the results. Some examples of validated instruments are the Epidemiology of Prolapse and Incontinence Questionnaire (EPIQ), the Pelvic Floor Distress Inventory Short Form (PFDI-20), and the International Consultation on Incontinence Questionnaire (ICIQ), etc.

In addition, it is important to clarify whether the approach will rely solely on clinical history to identify pelvic floor dysfunction, or whether both methods will be used. It is important to include validated methods in the data extraction, as this would provide a more robust basis for the results obtained. Therefore, it would be beneficial to include a specific field in the Excel file for data extraction to indicate how prevalence was measured in each study.

Finally, it is important to consider whether pelvic pain should be included or excluded from the analysis alongside pelvic floor dysfunction for a global analysis of pelvic floor dysfunction.

7. PLOS authors have the option to publish the peer review history of their article (what does this mean? ). If published, this will include your full peer review and any attached files.

**Do you want your identity to be public for this peer review?** For information about this choice, including consent withdrawal, please see our Privacy Policy .

Reviewer #1: **Yes: ** alireza hadizadeh

Reviewer #2: **Yes: ** Prof. Lilly Varghese

Reviewer #3: **Yes: ** Tokumasa Suemitsu M.D.

Reviewer #4: No

---

## [Author Response · Author response to Decision Letter 1]

19 Jan 2025

Response to Reviewers

Dear Editor,

Thank you for the opportunity to submit a revised version of the manuscript “Pelvic floor disorders and associated factors among women in sub-Saharan Africa: A Systematic Review and Meta-Analysis Protocol” for publication in PLOS ONE. We appreciate the time and effort that you and the reviewers dedicated to providing feedback on our manuscript and are grateful for the insightful comments on and valuable improvements to our paper. We have incorporated most of the suggestions made by the reviewers. Those changes appear as tracked changes within the manuscript. Please see below a point-by-point response to the reviewers’ comments and concerns.

Reviewers’ suggestions and comments to the Authors:

Reviewer #1: Reviewer comments:

The authors present a protocol for a systematic review of pelvic floor disorders and associated factors among women in sub-Saharan Africa. While this is an important topic for understanding the burden and determinants of pelvic floor disorders in this population, the rationale for publishing the protocol, as opposed to the actual study, is not clearly articulated. The protocol outlines a comprehensive approach to data collection and interpretation; however, the authors do not explain why the protocol itself warrants publication. Additionally, the aim and scope of this research might be better addressed through collaboration with the Global Burden of Disease study. Even if the authors wish to pursue this review by analyzing the literature, this approach does not necessarily justify the protocol's publication, as it neither significantly advances the literature nor provides a clear trajectory for future research. Moreover, the authors fail to present a comprehensive search strategy across relevant databases, which is crucial for conducting a systematic review. Given the stated goal of performing a thorough systematic search, it is imperative that the complete strategy be provided.

Response: Thank you for your valuable input, and we appreciate your observation. There are several advantages of publishing a review protocol, including informing the academic community about ongoing study and possibly preventing duplication. It also offers transparency and aids in the early detection and resolution of problems. We have uploaded the proposed search strategy as a supporting information, and we can modify it and will upload it along with the final manuscript.

Reviewer #2: Reviewer comments:

The review question is clear

The protocol is adequately developed based on international Rw guidelines

Tools such as Covidence helps for an efficient screening of the included studies

The methodological approach using PRISMA-P is adequate and clear as the review protocol.

How the research team will implement each step and who will be involved at each stage is well detailed

Joanna Bridge Institute's (JBI) appraisal should read as Joanna Briggs Institute.

Response: Thank you for your encouragement and insightful feedback. Joanna Bridge Institute's (JBI) was typing error and rephrased as Joanna Briggs Institute.

Reviewer #3: Reviewer comments:

Thank you for affording me the valuable opportunity to review this manuscript.

This methodological study offers significant insights into the prevalence and risk factors of pelvic floor disorders (PFDs) in sub-Saharan Africa, an area where existing evidence remains limited. The findings are poised to make a substantial contribution to the field.

However, the inclusion criteria need to be reassessed.

Pelvic floor disorders (PFDs) can be evaluated through clinical examination or self-administered questionnaires. For instance, the Pelvic Floor Disability Index (PFDI-20) is a self-reported questionnaire that does not determine prevalence but quantifies symptom burden.

In such cases, the findings primarily elucidate symptomatic presentation rather than true prevalence, as exemplified in the reference study (Suemitsu T, Mikuni K, Matsui H, Suzuki M, Takahashi T. Prevalence and Risk Factors of Pelvic Floor Disorders After Delivery in Japanese Women Using the Pelvic Floor Distress Inventory: A Retrospective Cohort Study. Cureus. 2023;15(6):e40152. Published 2023 Jun 8. doi:10.7759/cureus.40152).

The inclusion criteria will inevitably vary depending on whether the study aims to determine prevalence or symptom burden. Conducting a preliminary evaluation of previous studies to clarify their assessment methods, the volume of relevant literature, and the reliability of these sources would be prudent.

Additionally, I suggest considering treatment and intervention strategies, including rehabilitation, as secondary evaluation outcomes. Identifying risk factors does not always equate to their mitigation. For example, the increased incidence of PFDs associated with instrumental delivery, obstetric anal sphincter injuries (OASIS), and epidural analgesia is a well-documented phenomenon worldwide, not exclusive to sub-Saharan Africa.

A review of this nature may yield findings similar to previous studies. Nevertheless, presenting potential solutions would substantially enhance the review's significance. Therefore, I recommend incorporating this aspect into the evaluation criteria.

Response: Thank you for pointing this out. We have modified our inclusion criteria to “Observational studies that reported the prevalence of women with PFDs using validated data collection tools and conducted in sub-Saharan Africa …”. We have carried out a preliminary study to identify how studies defined PFD (whether through clinical examination or self-report). All of the studies we observed used one of the validated data collection tools, including the Pelvic Floor Distress Inventory Short Form (PFDI-20), Epidemiology of Prolapse and Incontinence Questionnaire (EPIQ), and the International Consultation on Incontinence Questionnaire (ICIQ). We will forward recommendations to stakeholders and interested organizations based on the review’s findings to contribute to the development of suitable care and preventative strategies.

Reviewer #4: Reviewer comments:

I would like to thank you for the opportunity to review the study protocol entitled "Pelvic Floor Disorders and Associated Factors Among Women in Sub-Saharan Africa: A Systematic Review and Meta-Analysis Protocol". This protocol raises considerations regarding the inclusion of studies for analysis, and I would like to make a few suggestions.

First, it is important to determine whether the studies selected for review have adequately addressed the prevalence of pelvic floor disorders. It is recommended that the included studies use validated assessment tools that are appropriate for the target population. The lack of validity may be a source of bias in the results. Some examples of validated instruments are the Epidemiology of Prolapse and Incontinence Questionnaire (EPIQ), the Pelvic Floor Distress Inventory Short Form (PFDI-20), and the International Consultation on Incontinence Questionnaire (ICIQ), etc.

In addition, it is important to clarify whether the approach will rely solely on clinical history to identify pelvic floor dysfunction, or whether both methods will be used. It is important to include validated methods in the data extraction, as this would provide a more robust basis for the results obtained. Therefore, it would be beneficial to include a specific field in the Excel file for data extraction to indicate how prevalence was measured in each study.

Finally, it is important to consider whether pelvic pain should be included or excluded from the analysis alongside pelvic floor dysfunction for a global analysis of pelvic floor dysfunction.

Response: we are grateful for your valuable feedback. Pelvic floor disorders (PFDs) can be evaluated through clinical examination or validated questionnaires. We will consider any study that can report the prevalence of women with PFDs using validated data collection tools and conducted in the SSA, though we have not identified a study that used clinical examination to decide PFD during our preliminary study. pelvic pain will not be assessed in the current study as an outcome variable.

---

## [Decision Letter · Decision Letter 1]

12 Feb 2025

Pelvic floor disorders and associated factors among women in sub-Saharan Africa: A Systematic Review and Meta-Analysis Protocol

PONE-D-24-38448R1

Dear Dr. Negera,

We’re pleased to inform you that your manuscript has been judged scientifically suitable for publication and will be formally accepted for publication once it meets all outstanding technical requirements.

Kind regards,

Richard Kao Lee, M.D.

Academic Editor

PLOS ONE

Additional Editor Comments (optional):

Reviewers' comments:

Reviewer's Responses to Questions

**Comments to the Author**

1. Does the manuscript provide a valid rationale for the proposed study, with clearly identified and justified research questions?

Reviewer #3: Yes

Reviewer #4: Yes

2. Is the protocol technically sound and planned in a manner that will lead to a meaningful outcome and allow testing the stated hypotheses?

Reviewer #3: Yes

Reviewer #4: Yes

3. Is the methodology feasible and described in sufficient detail to allow the work to be replicable?

Reviewer #3: Yes

Reviewer #4: Yes

4. Have the authors described where all data underlying the findings will be made available when the study is complete?

Reviewer #3: Yes

Reviewer #4: Yes

5. Is the manuscript presented in an intelligible fashion and written in standard English?

Reviewer #3: Yes

Reviewer #4: Yes

6. Review Comments to the Author

You may also provide optional suggestions and comments to authors that they might find helpful in planning their study.

Reviewer #3: Dear Authors,

I appreciate receiving your manuscript again.

I confirm you addressed the manuscript now and adequately improved the clarity and readability.

Now, I am sure that this manuscript will be accepted.

Reviewer #4: Thank you for addressing the pending comments - I feel that the manuscript has gained a lot of clarity and wish you success with your further studies.

7. PLOS authors have the option to publish the peer review history of their article (what does this mean? ). If published, this will include your full peer review and any attached files.

**Do you want your identity to be public for this peer review?** For information about this choice, including consent withdrawal, please see our Privacy Policy .

Reviewer #3: **Yes: ** Tokumasa Suemitsu

Reviewer #4: No

---

## [Editor Report · Acceptance letter]

PONE-D-24-38448R1

PLOS ONE

Dear Dr. Negera,

I'm pleased to inform you that your manuscript has been deemed suitable for publication in PLOS ONE. Congratulations! Your manuscript is now being handed over to our production team.

Kind regards,

on behalf of

Dr. Richard Kao Lee

Academic Editor

PLOS ONE